# Air-Frying Is a Better Thermal Processing Choice for Improving Antioxidant Properties of *Brassica* Vegetables

**DOI:** 10.3390/antiox12020490

**Published:** 2023-02-15

**Authors:** Ruchira Nandasiri, Breanne Semenko, Champa Wijekoon, Miyoung Suh

**Affiliations:** 1Department of Food & Human Nutritional Sciences, University of Manitoba, Winnipeg, MB R3T 2N2, Canada; 2Division of Neurodegenerative Disorders (DND), St. Boniface Hospital Albrechtsen Research Centre, 351 Tache Avenue, Winnipeg, MB R2H 2A6, Canada; 3Canadian Centre for Agri-Food Research in Health and Medicine (CCARM), St. Boniface Hospital Albrechtsen Research Centre, 351 Tache Avenue, Winnipeg, MB R2H 2A6, Canada; 4Agriculture Agri-Food Canada, Morden Research and Development Centre, Morden, MB T1J 4B1, Canada

**Keywords:** thermal processing, *Brassica* vegetables, kale, broccoli sprouts, air-frying, antioxidants

## Abstract

*Brassica* vegetables have demonstrated many health benefits over the years due to their composition of phenolic, flavonoid, and glucosinolate contents. However, these bioactive molecules can be easily depleted during gastronomic operations. Therefore, a sustainable method that improves their phenolic content and antioxidant activity is required for both the processors and consumers. Thermal processing has been demonstrated as a method to improve the phenolic content and antioxidant status of *Brassica* vegetables. In the current study, four different thermal processing methods, including freeze-drying, sautéing, steaming, and air-frying, were employed for five different *Brassica* vegetables, including kale, broccoli sprouts, Brussels sprouts, red cabbage, and green cabbage. The total phenolic content (TPC), total flavonoid content (TFC), and antioxidant activities were assessed using radical scavenging activity (DPPH and ABTS^•+^), reducing power (FRAP), and the chelating ability of metal ions. Among the methods tested, air-frying at 160 °C for 10 min showed the highest TPC, TFC, and antioxidant activity of the *Brassica* vegetables, while sautéing showed the lowest. The steam treatments were preferred over the freeze-drying treatments. Within the vegetables tested, both kale and broccoli sprouts contained higher antioxidant properties in most of the employed processing treatments. The results also indicated that there is a strong correlation between the TPC, TFC, and antioxidant activity (*p* < 0.05). This study indicates that air-frying could be used as a sustainable thermal processing method for improving biomolecules in *Brassica* vegetables.

## 1. Introduction

*Brassica* vegetables, particularly *Brassica oleraceae* (e.g., cabbage, Brussels sprouts, broccoli sprouts, kale), have gained attention over the years due to their numerous health benefits. Their associated health benefits include protective effects on type 2 diabetes, cardiovascular disease, coronary heart disease, and hypertension [1,2,3]. These vegetables are rich sources of fiber, vitamins, carotenoids, and minerals [2,4], including rich phenolic profiles with relatively higher antioxidant properties [4,5]. Among these natural antioxidants, flavonoids provide better protective properties as reducing agents and radical scavenging agents pertaining to antioxidant activity [5].

The common understanding is that when consumed raw, these *Brassica* vegetables would provide better nutritional benefits, whereas heat treatments and thermal processing decrease their nutrient content [5] with prolonged cooking. This controversial statement on thermal processing should be clarified with robust scientific evidence. The research question of whether thermal processing affects the phenolic composition and antioxidant activity of *Brassica* vegetables has arisen. Enhanced food-processing techniques are necessary to address the improved nutritional content in these gastronomic operations and fulfill the nutritional needs of the high-risk populations for chronic diseases [6]. 

Among the thermal processing techniques, pressurized steam, stir-frying, and air-frying have gained attention over the period [7,8,9,10,11]. In addition, freeze-drying has been popular due to its wide applicability in different food matrixes [12,13]. A recent study reported an improvement in the phenolic composition and antioxidant profiles of some *Brassica* vegetables, such as canola and mustard, using air-frying techniques compared to other thermal techniques [7,8]. Whether similar findings occur in *Brassica oleraceae* is of interest. Therefore, the current study was designed to assess the impact of different thermal processing techniques on the total phenolic content (TPC), total flavonoid content (TFC), and antioxidant activity of *Brassica oleraceae* vegetables. The findings of the current study would demonstrate the importance of thermal processing of *Brassica* vegetables as a method to improve their phenolic and flavonoid contents, ultimately improving health and the food industry. 

## 2. Materials and Methods

### 2.1. Materials

Five *Brassica* vegetables, including red cabbage, green cabbage, broccoli sprouts, Brussels sprouts, and kale, were selected for the current study based on their potential health benefits toward the positive outcomes of type 2 diabetes. The vegetables were selected from three different locations in Manitoba (south, central, and north) on the same date to obtain a representative sample. The grocery stores included Sobeys (south Winnipeg), Fresco (central Winnipeg), and Safeway (north Winnipeg). All of the vegetables were subjected to different processing conditions on the same day and stored at −80 °C until used for different assays. 

### 2.2. Chemicals

A Folin–Ciocalteu’s (FC) reagent, a total phenolic content (TPC) standard, iron (II) chloride hexahydrate (98%), iron (III) chloride hexahydrate (97%), iron (II) sulphate heptahydrate (99%), hydrogen chloride (HCl, 99%), sodium acetate, 2,4,6-tris-(2-pyridyl)-s-triazine (TPTZ > 98%), sinapic acid (>97%), 2,2-diphenyl-1-picrylhydrazyl (DPPH, 97%), and formic acid were all purchased from Fisher Scientific Canada Ltd. (Ottawa, ON, Canada). Aluminum chloride (AlCl_3_), sodium nitrite (NaNO_2_), sodium carbonate (Na_2_CO_3_), sodium hydroxide (NaOH), ferrozine, and disodium ethylenediaminetetraacetic acid (Na_2_EDTA) were purchased from Sigma Canada Ltd. (Mississauga, ON, Canada). Quercetin hydrate (>95%) and 2-amino-ethyl-diphenyl borate (98%) were purchased from Acros (Mississauga, ON, Canada). 

Extraction reagents, including methanol (optima grade) and ethanol (analytical grade), and standard compounds for high-performance liquid chromatography (HPLC) were purchased from Sigma Canada Ltd. (Mississauga, ON, Canada).

### 2.3. Application of Different Processing Techniques to Optimize the Phenolic Content

#### 2.3.1. Freeze-Drying Treatment

The freeze-drying of the vegetables was conducted based on the method described by Wu et al. [14], with slight modifications. Each type of vegetable was cut into pieces measuring 2 cm × 2 cm in size and stored for two hours at −80 °C prior to freeze-drying. The freeze-drying was conducted using a Labconco 4.5 Freezone freeze-dryer (Labconco Corporation, Kansas City, MO, USA) at a temperature of −50 °C for four days until the constant dry weight was recorded. After the freeze-drying, the vegetables were ground into fine particles and kept at −80 °C until further analysis (Figure 1). 

#### 2.3.2. Pressurized Steam Treatment 

The pressurized wet extraction of the vegetables was conducted using an instant pot (Instant Pot Duo Mini 3 Qt; model number: IPDUOMINI 3 Qt) at a temperature of 100 °C at 10.2 psi for 5 min. The vegetable-to-water ratio was kept at 5:1 according to the method described by Korus et al. [15], with a few modifications. After each steam treatment, the vegetables were drained and cut into pieces measuring 2 cm × 2 cm in size. The samples were freeze-dried according to the method described in Section 2.3.1 (Figure 1). 

#### 2.3.3. Air-Frying Treatment

The air-frying/roasting of the vegetables was conducted using the method described by Fadairo et al. [7,8], with slight modifications. The same instant pot (Instant Pot Duo Mini 3 Qt; model number: IPDUOMINI 3 Qt) was used here with the air-frying extension. All of the vegetables were subjected to an air-frying temperature and time combination of 160 °C for 10 min based on the optimized results of Fadairo et al. [7] and Nandasiri et al. [16]. The samples were cut into pieces measuring 2 cm × 2 cm in size after the air-frying, and they were freeze-dried according to the method described in Section 2.3.1 (Figure 1).

#### 2.3.4. Stir-Frying/Sautéing of the Vegetables

The same instant pot (Instant Pot Duo Mini 3 Qt; model number: IPDUOMINI 3 Qt) was used for the stir-frying operations using its in-built sauté function according to the method described by Nugrahedi et al. [17], with minor modifications. The vegetable-to-oil ratio of 10:1 was used for the stir-frying operations using canola oil (complements brand, Winnipeg, Manitoba). The stir-frying operation was conducted at 250 °C for 5 min. The vegetables were cut into pieces measuring 2 cm × 2 cm in size after the stir-frying, and they were freeze-dried as per the method described in Section 2.3.1 (Figure 1).

### 2.4. Sample Preparation

#### 2.4.1. Ultrasound-Assisted Extraction (UAE) of Phenolic Compounds 

The phenolic extraction was conducted using the methods described by Liu et al. [18] and Liang et al. [19], with slight modifications. In brief, 0.05 g of the freeze-dried vegetable sample was weighed and dissolved in 0.45 mL of 70% (*v/v*) methanol (at a solid-to-solvent ratio of 1:10). The phenolic extraction was carried out by ultrasound extraction using the SONOPLUS ultrasonic homogenizer HD 2200 system (BANDELIN electronic GmbH & Co. KG, Heinrichstraße, Berlin, Germany) with a power of 40% and a frequency of 20 kHz ± 500 Hz for 1 min at room temperature (25 °C). The extracts were centrifuged at 3000× *g* for 30 min at refrigeration conditions (4 °C) using the Eppendorf^™^ centrifuge 5804R (Fisher Scientific, Ottawa, ON, Canada). The extraction was repeated two more times, and the final volume was adjusted to 1.5 mL. The samples were concentrated using the Savant SPD 111V SpeedVac concentrator (Thermo Scientific, Mississauga, ON, Canada) for 5 h to remove the residual solvents, followed by freeze-drying at −50 °C for 2–3 h as per the method described in Section 2.3.1. The freeze-dried extracts were reconstituted in 0.5 mL of 100% (*v/v*) methanol and kept at −80 °C until further analysis. 

#### 2.4.2. Phenolic Extraction for Antioxidant Assays

The phenolic extractions for the antioxidant assays were conducted according to the method described by Singleton and Rossi [20], with a few modifications. In brief, 0.5 g of the freeze-dried vegetable powder was dissolved in 5.0 mL of 80% (*v/v*) methanol (at a sample-to-solvent ratio of 1:10) and kept at 25 °C for 15 h using an orbital shaker. After 15 h, the extracts were centrifuged at 3000× *g* for 30 min at refrigeration conditions (4 °C) using the Eppendorf^™^ centrifuge 5804R (Fisher Scientific, Ottawa, ON, Canada). The supernatant was collected and stored at −80 °C until further analysis. 

### 2.5. Antioxidant Activity of the Vegetables

#### 2.5.1. Assessment of Total Phenolic Content (TPC)

The total phenolic content (TPC) of the obtained extracts was estimated using the Folin–Ciocalteu method described by Thiyam et al. [21] and modified for a plate reader as described by Fadairo et al. [7]. In brief, 40 µL of the reconstituted plant extracts obtained from Section 2.4.2 were added to a Corning 9017 96-well microplate (Fisher Scientific, Ottawa, ON, Canada), followed by the addition of 120 µL of deionized water. A total of 40 µL of the FC reagent was added to the mixture and incubated for 5 min at 25 °C. After the incubation, 40 µL of Na_2_CO_3_ was added, and the sample mixture was kept in the dark for 1 h. Next, the absorbance was measured at 640 nm using a microplate reader (Bio-Tek Powerwave XS, New England, VT, USA). Methanol was substituted as the blank, and a TPC standard solution of 1000 mg/mL (Fisher Scientific, Mississauga, ON, Canada) was used to assemble the standard curve (Appendix A). The TPC was expressed as milligrams of gallic acid per gram of vegetable on a dry weight basis.

#### 2.5.2. Assessment of Total Flavonoid Content (TFC)

The total flavonoid content (TFC) of the vegetable extracts was determined by the aluminum chloride colorimetric method described by Zhishen et al. [22], with slight modifications. In brief, 25 µL of the reconstituted plant extracts obtained from Section 2.4.2 was added to a Corning 9017 96-well microplate (Fisher Scientific, Ottawa, ON, Canada), followed by the addition of 100 µL of deionized water (at a ratio of 1:4 (*v/v*)). The diluted sample was then mixed with 7.5 µL of 5% (*w/v*) NaNO_2_, and the reaction mixture was held at room temperature (25 °C) for 6 min. After, 7.5 µL of 10% (*w/v*) AlCl_3_ was added and held at room temperature (25 °C) for an additional 5 min. Then, 50 µL of NaOH (1 M) was added and mixed using the VWR™ analog mini vortex mixer (Henry Troemner LLC, Thorofare, NJ, USA). The absorbance was measured at 510 nm using a microplate reader (Bio-Tek Powerwave XS, New England, VT, USA). Methanol was substituted as the blank, and a quercetin standard solution of 1.0 mg/mL (Fisher Scientific, Mississauga, ON, Canada) was used to assemble the standard curve (Appendix A). The TFC was expressed as milligrams of quercetin per gram of vegetable on a dry weight basis.

#### 2.5.3. DPPH Free Radical Scavenging Assay

The DPPH radical scavenging activity of the extracted solution was measured using the DPPH assay described by Nandasiri et al. [23], with minor modifications. Briefly, 10 µL of the reconstituted plant extracts obtained from Section 2.4.2 was added to 290 µL of 100% (*v/v*) methanol in a Corning 9017 96-well microplate (Fisher Scientific, Ottawa, ON, Canada), followed by the addition of 10 µL of the prepared DPPH solution (0.05 mM). The samples were kept in the dark for 5 min to generate the radicals. The absorbance was measured at 516 nm using a microplate reader (Bio-Tek Powerwave XS, New England, VT, USA). Methanol was substituted as the blank. The free radical scavenging activity was measured using the following equation: $$\mathrm{Scavenging}\;\mathrm{Effect}\;\left(\%\right)=\frac{{(A}_{c}{-A}_{s})\;\times \;100}{A_{c}}$$
where A_c_ is the absorbance of the solvent control, and A_s_ is the absorbance of the sample.

#### 2.5.4. Ferric Reducing Antioxidant Power Assay (FRAP Assay)

Apart from the radical scavenging activity, the reducing power of the extracts was assessed using a modified method by Benzie and Strain [24]. The working reagent of FRAP was prepared by mixing acetate buffer (300 mM, pH = 3.6) and a TPTZ (2,4,6-tri [2-pyridyl]-s-triazine) solution (10 mM in 40 mM HCl) with a 20 mM FeCl_3_ solution at a ratio of 10:1:1 and keeping it at 37 °C until a straw-colored solution was formed. Briefly, 10 µL of the reconstituted plant extracts obtained from Section 2.4.2 was mixed with 90 µL of deionized water, followed by 90 µL of FRAP reagent in a Corning 9017 96-well microplate (Fisher Scientific, Ottawa, ON, Canada). The reaction mixture was then left in the dark for 8 min, and the absorbance was measured at 593 nm using a microplate reader (Bio-Tek Powerwave XS, New England, VT, USA). Deionized water was used as the blank, and a 1.0 mM solution of Trolox was used to create the standard curve (Appendix A).

#### 2.5.5. Ferrous Ion-Chelating Activity Assay Antioxidant Capacity

The chelating activity of the metals was assessed according to the method described by Dinis et al. [25], with a few modifications. In short, 10 µL of the reconstituted plant extracts obtained from Section 2.4.2 was added to a Corning 9017 96-well microplate (Fisher Scientific, Ottawa, ON, Canada), with 50 µL of a 2.0 mM FeCl_2_ solution and 20 µL of a 5.0 mM ferrozine solution prepared fresh daily. The total volume was adjusted to 280 µL using deionized water. The mixture was then kept at room temperature (25 °C) for 10 min, and the absorbance was measured at a wavelength of 562 nm using a microplate reader (Bio-Tek Powerwave XS, New England, VT, USA). Deionized water was used as the blank, and a 1.0 mM solution of Na_2_EDTA was used to create the standard curve (Appendix A).

#### 2.5.6. Total Antioxidant Capacity (TAC) Assay

The total antioxidant activity of each extract was assessed according to a protocol using a commercial KIT (Item # Cay709001-96; Cayman Chemicals, Ann Arbor, Michigan, USA) and a microplate reader (Bio-Tek Powerwave XS, Michigan, CA, USA). Deionized water was used as the blank, and a 1.0 mM solution of Trolox was used to create the standard curve (Appendix A).

## 3. Statistical Analysis

The results are presented as means ± standard deviations for all of the experiments that were conducted with three replicates. The normality of the data and the constant variance were confirmed prior to the statistical analyses [26]. The differences between the mean values of the main factor were determined by a two-way analysis of variance (ANOVA). A post-hoc analysis was conducted using the Tukey’s test, with 5% statistically significant differences (*p* < 0.05) considered statistically significant [26]. The SPSS statistical software version 26 (IBM, New York, NY, USA) was used to analyze the data.

## 4. Results and Discussion

### 4.1. Impact of Thermal Processing on Total Phenolic Content (TPC) and Total Flavonoid Content (TFC)

Phenolic compounds have been reported to contribute to plants’ flavor, color, and antioxidant activity. The TPC of the vegetables was determined using the Folin–Ciocalteu assay, which was based on the oxidation reaction of phenolic compounds in the presence of a mixture of phosphomolybdate and phosphotungstate [27]. The impact of the different thermal processing techniques was evaluated for the TPC. The results showed that air-frying, compared to other techniques, significantly (*p* < 0.05) increased the TPC of the vegetables, regardless of the variety (Figure 2). The statistical analysis indicated that the type of vegetable, the processing method, and the interactions between them were significant in the model statistics with an adjusted R^2^ value of 0.996 (Table 1). Previous studies by Nandasiri et al. [16] and Fadairo et al. [7,8] also showed improved TPC in canola and mustard by the application of air-frying. They observed the highest TPC values for the canola meal substrate conditions at 190 °C for 15 min (3.15 ± 0.14 mg GAE/g DW) and 20 min (3.05 ± 0.02 mg GAE/g DW), respectively. The present study optimized a condition with a lower temperature for a shorter time, 160 °C for 10 min, which exhibited much higher TPC values, ranging from 1.76 ± 0.11 mg GAE/g DW (green cabbage) to 5.87 ± 0.23 mg GAE/g DW (broccoli sprouts) (Figure 2). A study conducted by Ayaz et al. [28] reported that the TPC in kale leaves was around 1.37 mg/g on a fresh weight basis (FW), which was much lower compared to the values obtained by the current study.

The application of less time in gastronomic operations is often preferred for vegetables; hence, the air-frying condition of 160 °C for 10 min is a preferred option for *Brassica* vegetables. The air-fryer is designed to use hot air (~200 °C) to quickly cook foods with a continuous flow (using rapid air technology) circulating through the cooking chamber [29]. This rapid air technology creates an opportunity to create faster cooking operations, creating crispy coatings on the outside of the foods. Further, this rapid air technology reduces the preparation time (by 25–50%), pre-heating time (by 50–75%), and energy consumption (by 50%) [29]. No recent reports on the impact of air-frying of *Brassica* vegetables have been reported up to date, and this is the first study evaluating the impact of air-frying on their compositional changes. The application of higher temperatures (>140 °C) for shorter time intervals during the process of air-frying and the consistent circulation of hot air throughout the system make it ideal to preserve the nutrients and phenolic compounds without them leaching out [7]. Furthermore, in both canola and mustard, it was observed that certain thermo-generative phenolic compounds were also formed during the thermal process of air-frying, including canolol [7,8,16,30,31]. Canolol and other thermo-generative phenolic compounds demonstrated higher antioxidant potentials in their respective studies. Although not measured, the formation of these thermo-generative compounds could be associated with the higher TPC values and antioxidant activities obtained via the air-frying method in the present study.

In the present study, both the sauté and pressurized steam operations reported the lowest TPC levels (Figure 2). Further, the statistical analyses indicated that the type of vegetable, the processing method, and the interactions between them were significant in the model statistics with an adjusted R^2^ value of 0.983 (Table 1). The leaching of the nutrients, glucosinolates, and other phenolic compounds during the steaming process may lead to lower levels of TPC, regardless of the types of vegetables [32,33]. Paciulli et al. [32] reported that steamed Brussels sprouts contained a TPC of 0.25 ± 0.8 mg of GAE/g, which is comparable to the values reported in the current study using the pressured steam treatment by the instant pot, 0.19 ± 0.01 mg of GAE/g DW (Figure 2). A similar study conducted by Cieślik et al. [34] evaluated the impact of boiling, blanching, cooking, and freezing on different cruciferous vegetables, including Brussel sprouts, white and green cauliflower, broccoli, and curly kale, and observed considerable losses of total glucosinolates after blanching and cooking, with 30% and 72.4%, respectively, which affected the TPC levels. However, an interesting correlation trend was observed with the TPC and other antioxidant activities (Table 2). It was observed that the FRAP (0.936), TFC (0.863), and metal ion chelating activity (0.911) had a very high correlation with the TPC value (Table 2). However, a poor correlation was observed between the TPC and the DPPH radical activity (0.234) (Table 2).

Thermal processing strategies have been applied in the food industry since ancient times with the focus of delaying the inevitable deterioration of perishable foods between production and consumption [9]. Thermal processing, including steam, would destroy microbial pathogens while reducing the number of spoilage microorganisms and inactivating certain enzymes related to the relapse of foods [9]. Further, the use of oil in stir-frying operations might also have a detrimental effect on the TPC levels of the vegetables. During processing, some lipophilic phenolic compounds could leach out of the medium, resulting in losses in TPC values. A study conducted by Nugrahedi et al. [17] showed minimal differences in the quantity of glucosinolates among the treatment groups of different time–temperature combinations for Chinese cabbage. The authors reported that the inactivation of the myrosinase enzyme at higher temperatures of stir-frying would result in a negligible influence on its composition. However, in the current study, we observed low TPC values, ranging from 0.05 ± 0.00 mg GAE/g DW (green cabbage) to 0.35 ± 0.01 mg GAE/g DW (broccoli sprouts). In a different study, conducted on serrano peppers and jalapeno peppers, Mwebi and Ogendi [10] reported that the antioxidant concentration in the stir-fry, steamed, and boiled samples was nearly the same, but much higher than the raw samples. Interestingly, freeze-dried vegetables also had lower TPC levels compared to air-fried vegetables. In general, freeze-drying has been reported as a non-destructive method to preserve nutrients. Together with the above study [10], the current study demonstrated that freeze-drying operations were not an effective method compared to air-frying (Figure 2).

Certain phenolic compounds, including flavonoids, have also been proven to show strong antioxidant properties and health benefits [35,36]. The TFC of vegetables can be influenced by both intrinsic and extrinsic factors, including variety, maturity stage, cultivation location, and other processing conditions, such as temperature, pH, and pressure [37,38]. The current study found that the application of air-frying was able to significantly improve the TFC of vegetables, regardless of their varieties (Figure 3). For the air-fried treatment, kale contained the highest amount of TFC (90.76 ± 10.04 mg QE/g DW), followed by broccoli sprouts (67.21 ± 3.29 mg QE/g DW) and red cabbage (48.72 ± 2.01 mg QE/g DW). The TFC also had a similar correlation trend compared to the TPC (Table 2). The TFC demonstrated higher correlations among the FRAP (0.940), TPC (0.863), and metal ion chelating activity (0.939) (Table 2). However, similar to the TPC, a poor correlation was observed between the TFC and the DPPH radical activity (0.375) (Table 2).

These findings confirmed that air-frying is the preferred processing method for *Brassica* vegetables (Figure 3). It was reported that kale contains a relatively higher amount of flavonoids, including kaempferol, quercetin, isorhamntin, flavonol-3-O-glycosides, and flavonol-7-O-glycosides [39]. The higher TFC values could be associated with the distribution of these flavonoids. On the contrary, red cabbage contains a relatively higher amount of cyanidin compounds, which represents the comparable higher TFC values [40]. In addition, the processing conditions of air-frying, which involve a higher temperature for a shorter time, would further prevent them from leaching out of the vegetables during the processing. Interestingly, the sauté treatment demonstrated the lowest values for the TFC (Figure 3). The solubility of certain flavonoids in a hydrophobic medium is higher compared to an aqueous medium, and this could be associated with the lower TFC levels in the sauté treatment [41]. The formation of H-bonds with oil will further increase the solubility of the flavonoids in an oil medium, allowing the leaching out of the extractants and resulting in a lower flavonoid content in the final extracts of the sauté treatment [42]. Lemańska et al. [42] reported that both 3- and 5-hydroflavone formations with strong H-bonds with oxygen atoms from C4 = O inhibit deprotonationation and antioxidant potentials. However, with an increase in the pH, the shift would take place from the C3- and/or C5-hydroxyl groups to the C4 = carbonyl group, creating stable cations to promote electron donation for the flavonoid molecules. Interestingly, the steam treatments showed a higher TFC compared to the freeze-dry treatments (Figure 3). The shorter time exposure to heated vapor during the steam treatment would facilitate the breaking of the cell wall materials of these vegetables, thereby releasing the intracellular phenolic compounds into the medium and resulting in higher yields of TFC [43].

The current study confirms that thermal processing of *Brassica* vegetables for shorter durations at higher temperatures would enhance the extractability of flavonoids. These results were similar to those of Nandasiri et al. [23] from canola meal using accelerated solvent extraction. They found that there was a significant increase in the concentration of flavonoids among the extracts between the extraction temperatures of 140 °C (3.70 ± 0.11 µmol QE/g DM) and 180 °C (5.45 ± 0.27 µmol QE/g DM). The application of both pressure (1500 psi) and temperature had a favorable impact on the extractability of the flavonoids [23]. Similarly, Zago et al. [44] reported that both the pre-heating time and the pre-heating temperature had a positive effect on the extractability of the TFC on defatted hemp cake. Yet, the values were not significantly different. The TFC values ranged from 0.020 ± 0.01 µmol QE/g DM (160 °C; 15 min) to 0.23 ± 0.01 µmol QE/100 g DM (180 °C; 30 min) for the pre-treatment time/temperature combinations for the defatted hemp cake [44].

### 4.2. Impact of Thermal Processing on Antioxidant Activity

The antioxidant activity of the vegetable extracts was evaluated using different assays leading to different mechanisms, such as the radical scavenging activity, the chelating activity of metals, and the reducing power. Since the structures and active sites of phenolic compounds differ, each compound reacts differently, requiring different mechanisms of action to better understand their antioxidant activity.

#### 4.2.1. DPPH Free Radical Scavenging Activity of the *Brassica* Vegetables

The radical scavenging activity of the vegetable extracts was evaluated primarily using the DPPH radical scavenging activity. The DPPH radical scavenging activity is one of the most widely used colorimetric methods for measuring antioxidant activity using its scavenging capacity towards DPPH• radicals via an electron donating mechanism. In general, a higher antioxidant capacity would lead to a decrease in absorbance. These radicals could react in four different pathways, including proton-coupled electron transfer (PC-ET), electron transfer–proton transfer (ET-PT), sequential proton loss electron transfer (SPLET), and adduct formation (AF) [45,46]. Among the reported mechanisms, both the PC-ET and SPLET mechanisms are considered to follow the DPPH radical formation. Hence, for the 70% (*v/v*) methanol extractants with a dielectric constant of ε = 33, the SPLET mechanism is more applicable as it encourages ionization [45,47].

In the present study, both Brussels sprouts and broccoli sprouts had higher DPPH radical activity, despite the processing method, with an average radical scavenging activity of over 70% (Figure 4), indicating their higher antioxidant potential. The statistical analyses further indicated that the type of vegetable, the processing method, and the interactions between them were significant in the model statistics with an adjusted R^2^ value of 0.996 for the DPPH radical activity (Table 1). Further, it confirms that in the sprout stage, the composition of phenolics is much higher and more condensed compared to the mature stage. In addition, the minimal changes in the radical scavenging activity despite the processing operations indicate that the endogenous phenolic compounds present in both Brussels sprouts and broccoli sprouts are relatively stable and are minimally impacted by the gastronomic operations. In general, above 50% radical scavenging activity was reported for the air-fry, sauté, and steam operations, showing that thermal processing has favorable conditions toward antioxidant activity (Figure 4). A study conducted by Mwebi and Ogendi [10] also reported that the cooking operations have a different impact on the radical scavenging activity. The authors reported that the DPPH radical activity was in the order of microwaved > stir-fried > steamed > raw > boiled for both serrano peppers and jalapeno peppers. Further, in another study conducted by Turkmen et al. [48], it was reported that the radical scavenging activity of fresh vegetables was in the order of broccoli > pepper > spinach > green beans > peas > squash > leek. The authors also found a similar DPPH activity of 78.17% for fresh broccoli [48]. Similar to our findings, the authors also found a significant increase in the antioxidant activity in broccoli during the gastronomic operations of boiling (15.90%) and microwaving (16.68%) compared to its fresh form. The increments in antioxidant activity, specifically for *Brassica* vegetables, are reported to be due to the inactivation of peroxidases at higher processing temperatures, which reduces the pro-oxidant effects [48]. However, DPPH showed a lower correlation among TPC, TFC, and other antioxidant activities, demonstrating that the mechanism of action is different (Table 2).

#### 4.2.2. Ferric Reducing Antioxidant Power (FRAP) of the *Brassica* Vegetables

The antioxidant activity measured by reducing power demonstrated a different pattern from the DPPH radical scavenging activity. The results showed that the air-fry treatment facilitated the ferric reducing power of the vegetable extracts (Figure 5). For the air-fry treatment, both the broccoli sprouts (0.37 ± 0.00 mM TE/g DW) and kale (0.31 ± 0.01 mM TE/g DW) showed a higher FRAP value, while the green cabbage (0.15 ± 0.00 mM TE/g DW) showed the lowest (Figure 5). The statistical analyses also indicated that the type of vegetable, the processing method, and the interaction between them were significant in the model statistics with an adjusted R^2^ value of 0.998 for the reducing power (Table 1). An interesting higher correlation was observed between the FRAP activity and the TPC (0.936), TFC (0.940), and metal ion chelation (0.921) (Table 2). In addition, a moderate correlation was also observed between the FRAP and the total antioxidant activity (0.530) (Table 2). The trends between the TPC, TFC, and FRAP further illustrate the relationship between the phenolic content and its antioxidant activity.

The sauté treatment yielded the lowest antioxidant activity for all of the vegetable types. The lower FRAP values with the sauté treatment could be due to the leaching out of phenolic compounds into the oil fraction due to the formation of strong H-bonds [42]. The reducing power is closely linked with the electron-donating ability of a substance, which facilitates the transformation of ferric ions (Fe^3+^) (light brown) into ferrous ions (Fe^2+^) (blue) [23]. Consequently, in sautéing, due to the formation of H-bonds with the oil fraction, the electron-donating ability is further reduced, which results in a lower FRAP activity. Similar to both the TPC and TFC, the steam treatment demonstrated a higher FRAP value compared to the freeze-dry treatment (Figure 5). The application of heat in a pressurized environment for a shorter period would deactivate the myrosinase enzyme activity of the *Brassica* vegetables [49,50,51]. This phenomenon will activate the glucosinolates, thereby increasing the antioxidant activity of the thermally processed vegetables [51,52].

In addition, Gaspar et al. [53] explained the electron transfer mechanism of phenolic acids, demonstrating the association between the number of hydroxyl groups and the electrochemical potential of a phenolic acid. The authors showed that if a phenolic compound contains a higher number of hydroxyl groups, it will result in a lower electrochemical potential via *o*-quinone formation [53]. Teh et al. [54] also reported that higher extraction temperatures were correlated with a higher phenolic content and a higher antioxidant activity. The current study showed a similar trend for the TPC, TFC, and FRAP, agreeing with the findings of Teh et al. [54].

#### 4.2.3. Metal Ion Chelating Activity (MIC)

The chelating ability of the metal ions was assessed as a different mechanism of action for antioxidant activity. Similar to FRAP, the air-fried samples showed a comparatively higher antioxidant activity. Among the *Brassica* vegetables, kale showed the highest chelating activity for the air-fried samples (0.43 ± 0.03 mM EDTAE/g DW), while green cabbage showed the lowest (0.08 ± 0.04 mM EDTAE/g DW) (Figure 6). Sautéing showed the lowest chelating activity among the thermal processing treatments (Figure 6). The statistical analyses indicated that the type of vegetable, the processing method, and the interaction between them were significant in the model statistics with an adjusted R^2^ value of 0.974 for the metal ion chelating activity (Table 1). Similar to FRAP, the MIC demonstrated a higher correlation among the TPC (0.911), TFC (0.939), and FRAP (0.921) (Table 2). A moderate correlation was also observed between the total antioxidant activity (0.453) and DPPH (0.359) with MIC (Table 2). Mladěnka et al. [55] claimed that a neutral pH is favored for phenolic compounds to serve as metal ion chelators. They further stated that phenolic compounds containing 3-hydroxy-4-keto groups could create complexes, while phenolic compounds containing a catechol B ring are unable to chelate the metal ions [55]. It is speculated that in sautéing operations, more compounds containing a catechol B ring could be formed, resulting in lower metal ion chelating activity.

#### 4.2.4. Total Antioxidant Capacity

The total antioxidant capacity of the extracts was measured using ABTS^+^ scavenging activity commercial kits. The assay is based on a colorimetric principle evaluating the decay of ABTS^+^ in the presence of an antioxidant agent. For the current study, Trolox was used as the antioxidant agent. Dudonné et al. [56] illustrated a strong positive correlation among the TPC and ABTS^·+^ assay with an R^2^ value of 0.966. This was evident from the results of the current study, which showed similar trends for the TPC, TFC, FRAP, and metal ion chelation activity, with the air-frying treatment producing the highest antioxidant activity (Figure 7). The broccoli sprouts (0.96 ± 0.01 mM TE/g DW) showed the highest ABTS^+^ radical activity for the air-frying treatment, while both the green (0.43 ± 0.03 mM TE/g DW) and red (0.41 ± 0.03 mM TE/g DW) cabbage showed the lowest (Figure 7). The statistical analyses indicated that the type of vegetable, the processing method, and the interaction between them were significant in the model statistics. However, the adjusted R^2^ value for the total antioxidant activity was much lower compared to the other assays, with an adjusted R^2^ value of 0.528 (Table 1f). However, the correlation between the ABTS^+^ radical activity and DPPH was similar, with a moderate correlation among the TPC (0.541), TFC (0.464), FRAP (0.530), and MIC (0.453) (Table 2). Both the DPPH and ABTS^+^ radical activities showed no correlation (*r* = 0.013) (Table 2) for the current study. Even though both assays work on a radical scavenging mechanism, it shows that the mechanism of action between the two assays could be different.

Interestingly, both the freeze-dried and sautéed samples showed the lowest ABTS^·+^ radical activity for all of the *Brassica* vegetables. It was also found that the ABTS activity increases with the increasing polarity of the solvent [57]. Consequently, the application of sautéing would result in a low-polar medium, creating lower ABTS^·+^ radical activity (Figure 7).

## 5. Conclusions

The current study investigated the influence of different thermal processing methods to improve the antioxidant status of selected *Brassica* vegetables. The findings demonstrated that the application of air-frying improved the phenolic and flavonoid statuses and the antioxidant potential of the selected *Brassica* vegetables. Both kale and broccoli sprouts demonstrated the highest antioxidant activities during the air-frying treatment at 160 °C for 10 min. It was further observed that the antioxidant potential of the vegetables was improved with thermal processing. The pressurized steam treatment, which is the preferred gastronomic operation method, provided significantly (*p* < 0.05) lower antioxidant potentials compared to air-frying. Moreover, sautéing was the least favored thermal processing method, yielding lower phenolic and flavonoid contents and antioxidant activity. To the authors’ knowledge, this is the first study to evaluate the impact of air-frying on the phenolic and flavonoid contents and antioxidant activity of the selected *Brassica oleraceae* vegetables (kale, broccoli sprout, Brussels sprout, green cabbage, and red cabbage). The outcome of this study will further contribute to the food industry by introducing air-frying as an innovative and sustainable method to improve the antioxidant status of *Brassica* vegetables. This technique could be further applied to improve people’s nutritional statuses while creating new opportunities for producing functional vegetables.

## Figures and Tables

**Figure 1 antioxidants-12-00490-f001:**
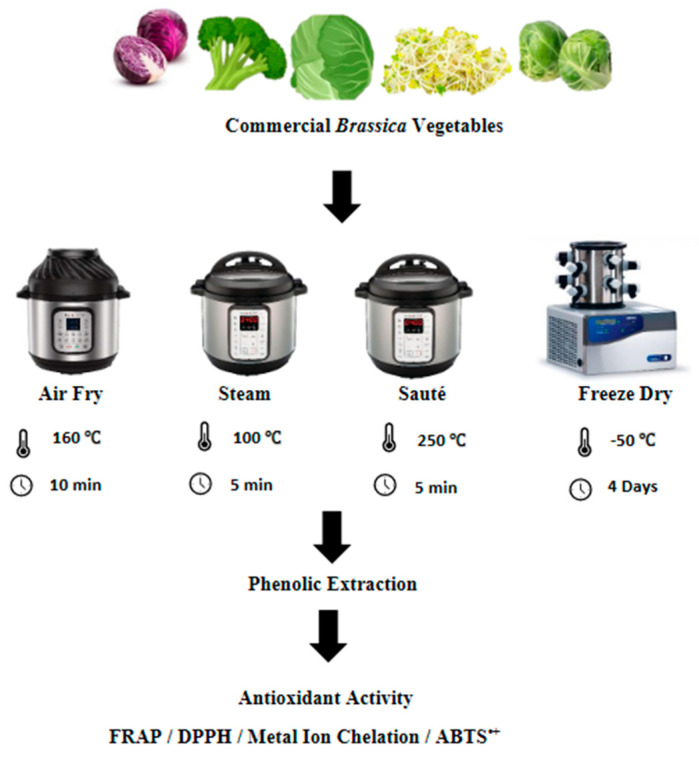
Summarized experimental approach for the processing methods. FRAP, ferric reducing antioxidant power; DPPH, 2,2-diphenyl-1-picrylhydrazyl; ABTS**^•+^**, 2,2′-azino-bis-3-ethylbenzothiazoline-6-sulfonic acid.

**Figure 2 antioxidants-12-00490-f002:**
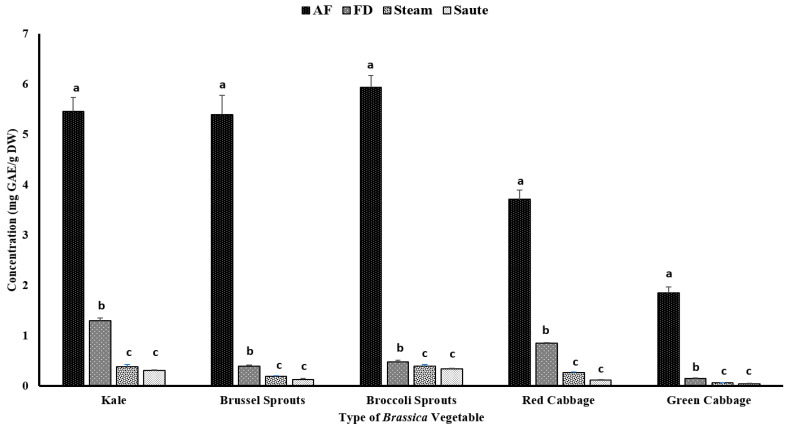
The effects of thermal processing techniques on the total phenolic content (TPC) of the selected *Brassica* vegetables. The bars represent means ± standard deviations (*n* = 3). The different letters for each vegetable indicate statistical differences based on a two-way analysis of variance. GAE, gallic acid equivalents; DW, dry weight; mg, milligram; g, gram; AF, air-fry; FD, freeze-dry.

**Figure 3 antioxidants-12-00490-f003:**
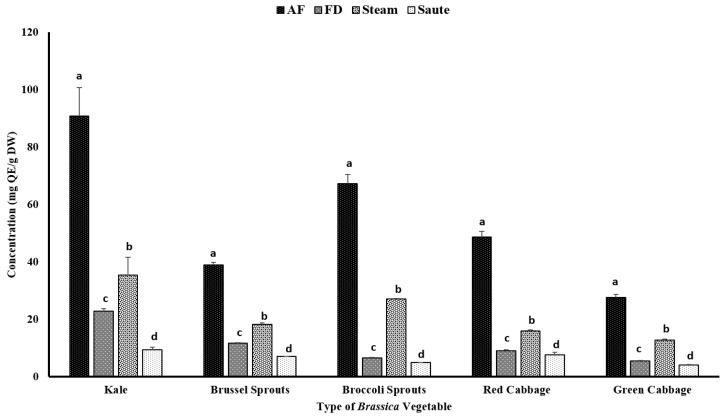
The effects of thermal processing techniques on the total flavonoid content (TFC) of the selected *Brassica* vegetables. The bars represent means ± standard deviations (*n* = 3). The different letters for each vegetable indicate statistical differences based on a two-way analysis of variance. QE, quercetin equivalents; DW, dry weight; mg, milligram; g, gram; AF, air-fry; FD, freeze-dry.

**Figure 4 antioxidants-12-00490-f004:**
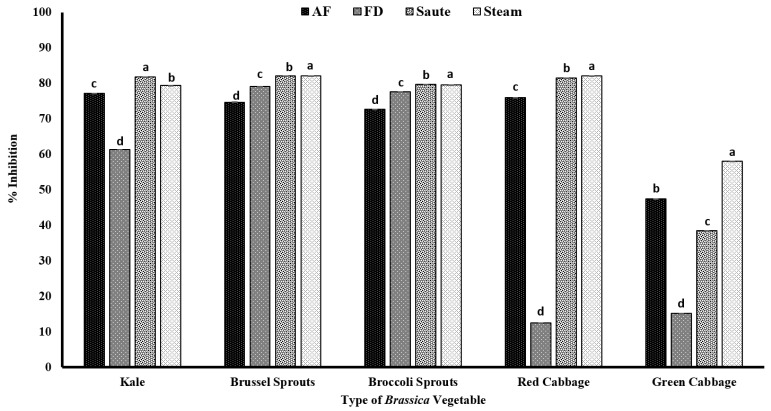
The effects of thermal processing techniques on the antioxidant activity of the selected *Brassica* vegetables measured by DPPH radical scavenging activity. The bars represent means ± standard deviations (*n* = 3). The different letters for each vegetable indicate statistical differences based on a two-way analysis of variance. AF, air-fry; FD, freeze-dry; DPPH, 2,2-diphenyl-1-picrylhydrazyl.

**Figure 5 antioxidants-12-00490-f005:**
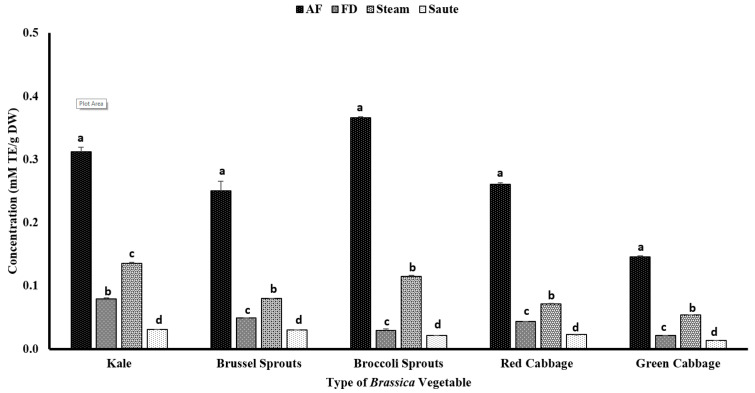
The effects of thermal processing techniques on the antioxidant activity of the selected *Brassica* vegetables measured by the FRAP antioxidant assay. The bars represent means ± standard deviations (*n* = 3). The different letters for each vegetable indicate statistical differences based on a two-way analysis of variance. mM, millimoles; g, gram; TE, Trolox equivalent; AF, air-fry; FD, freeze-dry; FRAP, ferric reducing antioxidant power.

**Figure 6 antioxidants-12-00490-f006:**
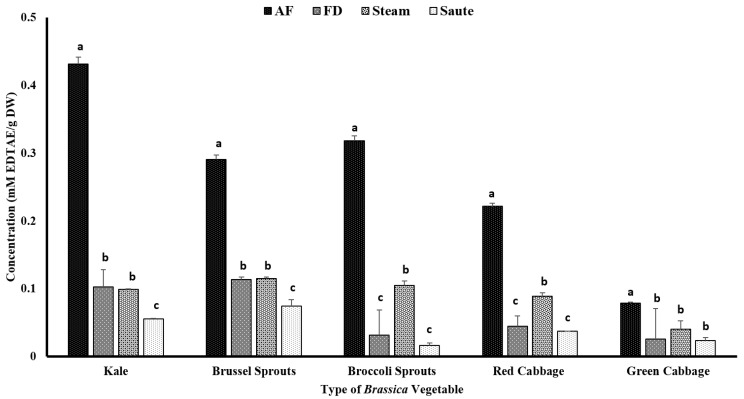
The effects of thermal processing techniques on the antioxidant activity of the selected *Brassica* vegetables measured by the chelating ability of the metals. The bars represent means ± standard deviations (*n* = 3). The different letters for each vegetable indicate statistical differences based on a two-way analysis of variance. mM, millimoles; g, gram; AF, air-fry; FD, freeze-dry; EDTA, ethylenediaminetetraacetic acid.

**Figure 7 antioxidants-12-00490-f007:**
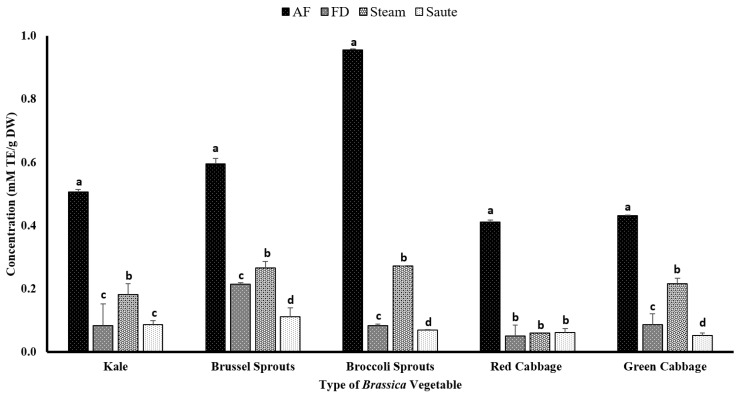
The effects of thermal processing techniques on the total antioxidant capacity of the selected *Brassica* vegetables measured by the ABTS radical scavenging assay. The bars represent means ± standard deviations (*n* = 3). The different letters for each vegetable indicate statistical differences based on a two-way analysis of variance. mM, millimoles; g, gram; TE, Trolox equivalent; AF, air-fry; FD, freeze-dry; ABTS, 2,2′-azino-bis-3-ethylbenzothiazoline-6-sulfonic acid.

**Table 1 antioxidants-12-00490-t001:** Impact of thermal treatments on the total phenolic (a) and total flavonoid (b) contents and different antioxidant activities (c–f) of the selected *Brassica* vegetables.

Source	Type III Sum of Squares	df	Mean Square	F	Sig.	Observed Power
**a: Total phenolic content (TPC)**						
Corrected Model	221.145	19	11.639	699.000	<0.001	1.000
Intercept	110.157	1	110.157	6615.546	<0.001	1.000
Veg	14.429	4	3.607	216.633	<0.001	1.000
Treatment	181.493	3	60.498	3633.225	<0.001	1.000
Veg × Treatment	25.223	12	2.102	126.232	<0.001	1.000
Error	0.666	40	0.017			
Total	331.969	60				
Corrected Total	221.811	59				
R^2^ = 0.997 (Adjusted R^2^ = 0.996); level of significance: 0.05						
Corrected model for TPC = Intercept + Veg + Treatment + Veg × Treatment						
**b: Total flavonoid content (TFC)**						
Corrected Model	29,157.237	19	1534.591	169.805	<0.001	1.000
Intercept	29,577.335	1	29,577.335	3272.772	<0.001	1.000
Veg	4207.408	4	1051.852	116.389	<0.001	1.000
Treatment	19,693.546	3	6564.515	726.373	<0.001	1.000
Veg × Treatment	3678.639	12	306.553	33.921	<0.001	1.000
Error	316.309	35	9.037			
Total	63,114.012	55				
Corrected Total	29,473.545	54				
R^2^ = 0.989 (Adjusted R^2^ = 0.983); level of significance: 0.05						
Corrected model for TFC = Intercept + Veg + Treatment + Veg × Treatment						
**c: Ferric reducing antioxidant power (FRAP)**						
Corrected Model	0.654	19	0.034	1475.981	<0.001	1.000
Intercept	0.668	1	0.668	28,620.643	<0.001	1.000
Veg	0.051	4	0.013	549.750	<0.001	1.000
Treatment	0.555	3	0.185	7926.548	<0.001	1.000
Veg × Treatment	0.048	12	0.004	172.083	<0.001	1.000
Error	0.001	40	2.333 × 10^−5^			
Total	1.323	60				
Corrected Total	0.655	59				
R^2^ = 0.999 (Adjusted R^2^ = 0.998); level of significance: 0.05						
Corrected model for FRAP = Intercept + Veg + Treatment + Veg × Treatment						
**d. 2,2-diphenyl-1-picrylhydrazyl (DPPH) activity**						
Corrected Model	4.075	19	0.214	825.807	0.000	1.000
Intercept	21.799	1	21.799	83,943.365	0.000	1.000
Treatment	2.676	3	0.892	3434.287	0.000	1.000
Veg	0.859	4	0.215	826.851	0.000	1.000
Treatment × Veg	0.540	12	0.045	173.339	0.000	1.000
Error	0.010	40	0.000			
Total	25.884	60				
Corrected Total	4.085	59				
R^2^ = 0.997 (Adjusted R^2^ = 0.996); level of significance: 0.05						
Corrected model for DPPH = Intercept + Veg + Treatment + Veg × Treatment						
**e. Metal ion chelation (MIC) activity**						
Corrected Model	0.551	19	0.029	97.496	0.000	1.000
Intercept	0.636	1	0.636	2138.872	0.000	1.000
Treatment	0.375	3	0.125	420.579	0.000	1.000
Veg	0.099	4	0.025	83.451	0.000	1.000
Treatment × Veg	0.099	12	0.008	27.861	0.000	1.000
Error	0.009	30	0.000			
Total	1.167	50				
Corrected Total	0.559	49				
R^2^ = 0.984 (Adjusted R^2^ = 0.974); level of significance: 0.05						
Corrected model for MIC = Intercept + Veg + Treatment + Veg × Treatment						
**f.** **2,2′-azino-bis-3-ethylbenzothiazoline-6-sulfonic acid (ABTS^•+^) activity**						
Corrected Model	6.078	19	0.320	4.119	0.000	0.999
Intercept	1.912	1	1.912	24.617	0.000	0.998
Treatment	2.589	3	0.863	11.112	0.000	0.998
Veg	1.297	4	0.324	4.175	0.007	0.880
Treatment × Veg	2.176	12	0.181	2.335	0.026	0.887
Error	2.640	34	0.078			
Total	10.252	54				
Corrected Total	8.718	53				
R^2^ = 0.697 (Adjusted R^2^ = 0.528); level of significance: 0.05						
Corrected model for ABTS = Intercept + Veg + Treatment + Veg × Treatment						

**Table 2 antioxidants-12-00490-t002:** Pearson correlation analyses between different antioxidant activity, total phenolic content, and total flavonoid content.

	TPC	FRAP	TFC	DPPH	MIC	ABTS
TPC	1					
FRAP	0.936 **	1				
TFC	0.863 **	0.940 **	1			
DPPH	0.234	0.34 8 **	0.375 **	1		
MIC	0.911 **	0.921 **	0.939 **	0.359 *	1	
ABTS	0.541 **	0.530 **	0.464 **	0.013	0.453 **	1

** Correlation is significant at the 0.01 level; * Correlation is significant at the 0.05 level; TPC, total phenolic content; TFC, total flavonoid content; FRAP, ferric reducing antioxidant power; DPPH, 2,2-diphenyl-1-picrylhydrazyl; ABTS**^•+^**, 2,2′-azino-bis-3-ethylbenzothiazoline-6-sulfonic acid; MIC, metal ion chelation activity.

## Data Availability

Data will be available on request.

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
