# Peer review of "Air-Frying Is a Better Thermal Processing Choice for Improving Antioxidant Properties of Brassica Vegetables"

_antioxidants, 2023, doi:10.3390/antiox12020490_

Round 1

Reviewer 1 Report

The authors of the manuscript, "Thermal Processing via Air Frying Improves the Antioxidant Properties of Brassica Vegetables," evaluated the effects of using different heat treatment techniques on the polyphenol content and antioxidant activity of five brassica vegetable species. The study design itself may not be very original, but the manuscript provides very practical knowledge that can be applied to the food industry. The topics of the manuscript are in line with the scope of the journal. I have only a few minor comments:

1). For what reason did the authors make the extracts by two methods (sections 2.4.1 and 2.4.2)?

2) Figures should be corrected (missing or misplaced significance of differences).

Author Response

The authors of the manuscript, "Thermal Processing via Air Frying Improves the Antioxidant Properties of Brassica Vegetables," evaluated the effects of using different heat treatment techniques on the polyphenol content and antioxidant activity of five brassica vegetable species. The study design itself may not be very original, but the manuscript provides very practical knowledge that can be applied to the food industry. The topics of the manuscript are in line with the scope of the journal. I have only a few minor comments:

Thank you for your positive feedback on the manuscript.  We are pleased to share with the readers with this new finding on the air-frying improving antioxidant activity of the Brassica Vegetables.

1). For what reason did the authors make the extracts by two methods (sections 2.4.1 and 2.4.2)?

The extraction methods employed in this study are most common for the extraction of the antioxidant activity of Brassica vegetables. According to literature, extraction of phenolic compounds of the Brassica vegetables could be optimized using the ultrasound aided extractions.

2) Figures should be corrected (missing or misplaced significance of differences).

Thank you for your comment. We have updated all the figures with the misplaced significant differences.

Reviewer 2 Report

Dear Editor and Authors,

The manuscript ID: antioxidants-2220017 entitled Thermal Processing via Air Frying Improves the Antioxidant Properties of Brassica Vegetables is interesting, offering valuable information regarding effects of four different thermal processing methods, including freeze drying, sauteing, steam and air frying on antioxidants properties of five different Brassica vegetables, including kale, broccoli sprout, brussels sprout, red cabbage, and green cabbage. However, the following are the minor comments.

Lines 26, 105, 114, 122, 133, 134, 146, 156, 170, 171, 184, 199, 209, 238, 239, 241:  min instead of minutes; please verify in all the manuscript!

Line 223: p<0.05 instead of p>0.05

 Five replicates were specified in Statistical analysis section, whereas under the Figure 2 is mentioned n=3. Please verify?

 Figure 2: please accurate insert the letters for significance and explain the symbols used for abbreviations.

Table 1: please verify the table title!

Please insert in the Table 1 the predictive models obtained for total phenolic content, total flavonoid content, Ferric reducing antioxidant power, 2,2-diphenyl-1-picrylhydrazyl (DPPH) activity, metal ion chelation (MIC) activity, and 2, 2'-azino-bis-3-ethylbenzothiazoline-6-sulfonic acid (ABTS•+) activity.

 Figures 3, 5: please insert the letters for significance

Figures 4, 6: please verify the letters position!

Line 507-508: … lowest correlation with a r = 0.013. Please reformulate! In this case, the correlation coefficient between DPPH and ABTS·+ radical activity is non-significant at p < 0.05.

Author Response

The manuscript ID: antioxidants-2220017 entitled Thermal Processing via Air Frying Improves the Antioxidant Properties of Brassica Vegetables is interesting, offering valuable information regarding effects of four different thermal processing methods, including freeze drying, sauteing, steam and air frying on antioxidants properties of five different Brassica vegetables, including kale, broccoli sprout, brussels sprout, red cabbage, and green cabbage. The main purpose of this work was to evaluate the impact of different thermal processing techniques on the total phenolic content (TPC), total flavonoid content (TFC), and antioxidant activity on Brassica oleraceae vegetables. However, the following are the minor comments.

Thank you for your positive feedback on our manuscript.

Lines 26, 105, 114, 122, 133, 134, 146, 156, 170, 171, 184, 199, 209, 238, 239, 241:  min instead of minutes; please verify in all the manuscript!

We have updated the document replacing minutes with min

Line 223: p<0.05 instead of p>0.05

The correction has been made.

 Five replicates were specified in Statistical analysis section, whereas under the Figure 2 is mentioned n=3. Please verify?

 The replicate number was 3 for each analysis, which is corrected in this submission.

 Figure 2: please accurate insert the letters for significance and explain the symbols used for abbreviations.

We have updated all the figures. All the abbreviations were explained in the parenthesis.

Table 1: please verify the table title!

Thank you for pointing out this. We have updated the table title to

“Impact of thermal treatments on total phenolic (a) and total flavonoid (b) content different antioxidant activities (c, d, e, f) of the selected Brassica vegetables”

Please insert in the Table 1 the predictive models obtained for total phenolic content, total flavonoid content, Ferric reducing antioxidant power, 2,2-diphenyl-1-picrylhydrazyl (DPPH) activity, metal ion chelation (MIC) activity, and 2, 2'-azino-bis-3-ethylbenzothiazoline-6-sulfonic acid (ABTS•+) activity.

Thank you for your suggestion. We have updated the table with the model statistics for all the parameters.

 Figures 3, 5: please insert the letters for significance

 All figures have been updated.

Figures 4, 6: please verify the letters position!

We have updated all the figures.

Line 507-508: … lowest correlation with a r = 0.013. Please reformulate! In this case, the correlation coefficient between DPPH and ABTS·+ radical activity is non-significant at p < 0.05.

Thank you for pointing this out. We have updated the sentence to

“Both DPPH and ABTS·+ radical activity showed no correlation (r = 0.013) (Table 2) for the current study.”

Reviewer 3 Report

The main purpose of this work was to evaluate the impact of different thermal processing techniques on the total phenolic content (TPC), total flavonoid content (TFC), and antioxidant activity on Brassica oleraceae vegetables.

In general, this manuscript is well-written and presented relevant results about thermal processing of Brassica oleraceae vegetables. Since, it presents a complete study of the influence of thermal processing techniques on the antioxidant properties of these vegetables.

The methodology is well specified and referenced, there is no improvement to be added.

The conclusions are consistent with the evidence and arguments presented in this study.

Two specific comments below:

Page 1, Title:

It does not represent all the research done.

It is important to note that the air frying was not the only thermal processing studied. Therefore, the title could be changed for Impact of Different Thermal Processing on the Antioxidant Properties of Brassica Vegetables.

Pages 6-8, Table 1: The content of this table needs to be better presented.

Author Response

Thank you for reviewing the manuscript. We have addressed all the concerns raised in the response to the review comments section.
